# Analysis of Orbital Atmospheric Density from QQ-Satellite Precision Orbits Based on GNSS Observations

Yueqiang Sun [1,2,3,4], Bowen Wang [1,2,3,4], Xiangguang Meng [1,2,3,4,*], Xinchun Tang [5], Feng Yan [5], Xianguo Zhang [1,2,3,4], Weihua Bai [1,2,3,4], Qifei Du [1,2,3,4], Xianyi Wang [1,2,3,4], Yuerong Cai [1,2,3,4], Bibo Guo [5], Shilong Wei [5], Hao Qiao [1,3,4], Peng Hu [1,2,3,4], Yongping Li [1,2,3,4] and Xinyue Wang [1,2,3,4]

1   National Space Science Center, Chinese Academy of Sciences, Beijing 100190, China
2   University of Chinese Academy of Sciences, Beijing 100190, China
3   Beijing Key Laboratory of Space Environment Exploration, National Space Science Center, Beijing 100190, China
4   Key Laboratory of Science and Technology on Space Environmental Situation Awareness, Chinese Academy of Sciences, Beijing 100190, China
5   Shenzhen Aerospace Dongfanghong Satellite Ltd., Shenzhen 518057, China
*   Correspondence: xgmeng@nssc.ac.cn

**Abstract:** Atmospheric drag provides an indirect approach for evaluating atmospheric mass density, which can be derived from the Precise Orbit Determination (POD) of Low Earth Orbit (LEO) satellites. A method was developed to estimate nongravitational acceleration, which includes the drag acceleration of the thermospheric density model and empirical force acceleration in the velocity direction from the centimeter-level reduced-dynamic POD. The main research achievements include the study of atmospheric responses to geomagnetic storms, especially after the launch of the spherical Qiu Qiu (QQ)-Satellite (QQ-Satellite) with the global navigation system satellite (GNSS) receiver onboard tracking the Global Positioning System (GPS) and Beidou System (BDS) data. Using this derivation method, the high-accuracy POD atmospheric density was determined from these data, resulting in better agreement among the QQ-Satellite-derived densities and the NRLMSISE-00 model densities. In addition, the POD-derived density exhibited a more sensitive response to magnetic storms. Improved accuracy of short-term orbit predictions using derived density was one of the aims of this study. Preliminary experiments using densities derived from the QQ-Satellite showed promising and encouraging results in reducing orbit propagation errors within 24 h, especially during periods of geomagnetic activity.

**Keywords:** QQ-Satellite; atmospheric mass density; precise orbit; drag acceleration; propagation

## 1. Introduction

The accurate and reliable estimation and prediction of the trajectories of Low Earth Orbit (LEO) satellites represent challenging tasks for researchers in the field of astrodynamics. To date, the inaccuracy of existing empirical atmospheric mass density models is one of the main challenges [1]. In the LEO orbit altitude range of 200–2000 km, the most significant disturbance force that acts on LEO satellites is typically atmospheric drag. Because atmospheric density depends on external drivers, especially solar and geomagnetic activities, this force is highly dynamic, thus complicating drag modeling [2].

During the main phase of geomagnetic activities, the field-aligned ring currents increase the total amount of energy in the ionosphere–thermosphere system at high latitudes, which is driven by a large amount of magnetospheric energy [3,4]. Then, the lower atmosphere inflates upward, and LEO satellites experience enhanced orbital drag effects, which consequently subject the orbital altitude decay of satellites to further increased atmospheric drag forces. These drag effects ultimately introduce large biases and errors in satellite trajectory tracking, which increase as space weather activities become more frequent and

intense [1]. For LEO space objects, the orbit prediction errors over a few days can reach tens of kilometers, potentially causing various problems in space applications [5].

The accuracy of the widely utilized atmospheric empirical models, including the Exponential, Jacchia-Bowman 2008 [6], DTM 2000 [7], and NRLMSISE-00 [8] models, is generally considered to be 15% [9]. The accuracy of these empirical density models is limited because they do not precisely model the perturbations and dynamics variations of the atmosphere but use a fixed set of mathematical representations to describe and define the general behaviors of observations and measurements [10]. The remaining errors are strongly related to the initial design shape of the spacecraft orbit, the orbital perigee, and the apogee altitude, which also experience decreasing orbital altitude as the atmospheric density increases [11].

In the last decade, the application of calibration methods has resulted in significant decreases in the density of modeling errors. In particular, the high-accuracy satellite drag model (HASDM),which uses the observations of tens of objects in the Space Surveillance Network for dynamic calibration, can significantly improve the estimated thermospheric densities [12]. Moreover, simultaneous observations of satellite drag can be estimated using the selected correction parameters. In contrast to the HASDM, the trajectory and other tracking data are freely available to the scientific community, and the density correction is estimated separately from the satellite trajectory [13].

The results of existing models are far from satisfactory and developing a thermospheric mass density model with high accuracy remains a challenging task. The decisive requirements for the development of such mass density models mainly include indirect and direct observation methods with adequate temporal and spatial resolutions as well as a higher overall quality [5]. Although the Challenging Minisatellite Payload (CHAMP) [14], Gravity Field and Steady-State Ocean Circulation Explore (GOCE), and Gravity Recovery and Climate Experiment (GRACE) [15] missions have produced high-quality mass density measurements from onboard accelerometers, the acceleration due to nongravitational forces acting on the satellite can be accurately captured by such instruments. However, more convenient and extensive spatial and temporal coverage can be obtained from the General Perturbation and Precise Orbit Determination (POD) datasets of LEO objects by using the semiaxis decay methods of Two-Line Elements (TLE) [16] or precise orbit ephemerides [17], respectively. In addition, drag perturbation acceleration is a proper solution [18]. Since the early 1960s, orbit prediction and POD software have been widely available for investigating in depth various space environments, such as simulating and forecasting space weather activities.

The method of retrieving the total mass densities used in this research is based upon the fundamentals of the nongravitational drag acceleration from orbital observations, which is a more accurate approach. The advanced development of POD has become a regular routine in satellite tracking and orbit determination for more LEO satellites, and the accuracy of such orbital element results is better than that from orbit ephemeride observations. Studies conducted [19] using Global Navigation System Satellite (GNSS) observed nongravitational accelerations can serve as a baseline for correcting Swarm-C accelerometer data. The acceleration of all Swarm satellites is converted into the atmospheric mass density, but the geometry of satellites is difficult to simulate and may cause bias. Montenbruck et al. [20] investigated the reconstruction of the empirical acceleration of the GRACE-B satellite by using the estimation method of batch least processing and the Kalman filter. Their results indicated that the overall variation in the empirical acceleration was comparable between these two techniques, but the acceleration magnitudes based on these two approaches showed multiplicative deviations.

To date, tens of spherical satellites have decayed or are in orbit, including Starshine, Orbital Debris Radar Calibration Sphere 2 (ODERACS), Geo Forschungs Zentrum Potsdam 1 (GFZ-1), Stella, QSAT, Calspheres, and the Russian Taifun. These spherical satellites are particularly well suited for estimating the average densities of the upper atmosphere because their structure consists of mirrored spheres. The ballistic coefficients can be

obtained by numerical calculations, which are substantially unrelated to the orientation with respect to the direction of motion. Therefore, the ODERACS orbital data are used to evaluate the deviations of the MSIS-90 and Jacchia-71 density models [21]. All three Starshine satellites have been used to retrieve the thermospheric mass density from the orbital elements, and the absolute uncertainties of the derived densities as a function of time are within ±6% [22]. The QSAT sphere microsatellite uses a correction–prediction strategy to revise the atmospheric density model, and the best improvement in the 24 h orbit prediction precision is approximately 171 m [23].

The QQ-Satellite (MD-1) was cooperatively designed and developed by the National Space Science Center, CAS, and Shenzhen Aerospace Dongfanghong Satellite, Ltd. (Shenzhen, China). The QQ-Satellite is China's new spherical satellite and was launched in the second half of 2021, as shown in Figure 1. The QQ-Satellite was designed to research in depth the response mechanism and characteristics of the atmospheric environment under the disturbance of solar and geomagnetic activities and to provide parameters that can improve empirical atmospheric models. To achieve the above scientific goals, the QQ-Satellite is equipped with the main payload Orbiting Atmospheric Sounder, which includes a high-precision GNSS receiver GPOD, an integrated atmospheric density sensor, and other units.

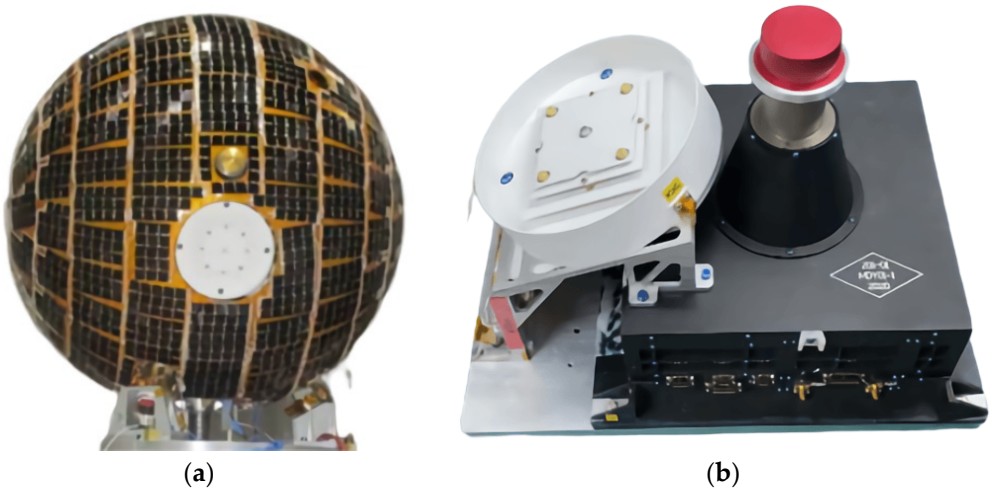

(**a**)                                                                  (**b**)

**Figure 1.** (**a**) QQ-Satellite structure; (**b**) High-precision GNSS receiver GPOD payload layout.

Satellites equipped with a high-precision GNSS receiver, such as the QQ-Satellite, can provide precise nongravitational accelerations and orbit ephemerides with an accuracy of a few centimeters [24]. Thus, an accurate atmospheric mass density can be extracted from the POD process, and the effectiveness and accuracy of orbit prediction can be validated [25]. McLaughlin et al. [26] used a technique of retrieving mass densities from the CHAMP satellite precise orbits ephemeris, which reflected the accelerometer-derived density variations along the CHAMP trajectory. However, these values were restricted to a temporal resolution of minutes.

One aspiration of this study was to investigate and explore the preciseness of the orbital nongravitational acceleration of the QQ-Satellite in deriving accurate thermospheric mass densities to improve the orbital propagation accuracy. The paramount interests include the ability to penetrate the reaction of the upper atmosphere to geomagnetic storms and the consequent increase in orbital drag as well as the capability of accurately predicting subsequent satellite orbits.

In this study, we used the precise orbit and drag perturbation acceleration of the QQ-Satellite to estimate atmospheric mass densities with an acute temporal resolution. Special attention was given to the performance of the model specifically during extreme geomagnetic storms. The drag perturbation equation of orbital drag acceleration is the baseline of the method used in this study. A comprehensive discussion of the requirements

is provided as follows. The density results were compared to the atmospheric density models, which also reveal the intra-diurnal temporal and spatial (latitudinal and longitudinal) variations. In addition, the differences between the POD-derived densities and Naval Research Laboratory Mass Spectrometer and Incoherent Scatter Radar Exosphere (NRLMSISE-00) densities, as well as the solar and geomagnetic indices, were evaluated. The QQ-Satellite derived density was used to calibrate the NRLMSISE-00 empirical model, which improved the accuracy of orbit propagation. This application demonstrates the usefulness of the approach. Finally, a discussion and conclusions are provided at the end of this article.

## 2. Materials and Methods

### 2.1. Data Source

The QQ-Satellite is dedicated to the in situ detection of the atmospheric composition and thermospheric density, obtaining the temporal and spatial evolution principle of the orbital atmosphere. The QQ-Satellite mission has generated volumes of data, including precise orbit positions and in situ detection data of the atmospheric density. In situ POD detection data from 1 to 8 November 2021 were considered due to high geomagnetic activity during this period. The detailed geomagnetic index is provided in the next section.

The QQ-Satellite was sent to a sun-synchronous orbit (SSO) with an orbital inclination of 97.46° and an altitude of 520 km, nearly 95 min around the Earth, via the Long March CZ-2D Y53 rocket on 14 October 2021. The coverage of the local-time latitude of the QQ-Satellite is shown in Figure 2. The dawn-dusk SSO enables the local-time drifts of the QQ-Satellite to be very small and provides favorable conditions for studying medium- and long-term variations in atmospheric mass density. In fact, the considerable coverage of latitudes and local times for this SSO is conducive to the advancement of atmospheric models when the QQ-Satellite-derived density observations are assimilated.

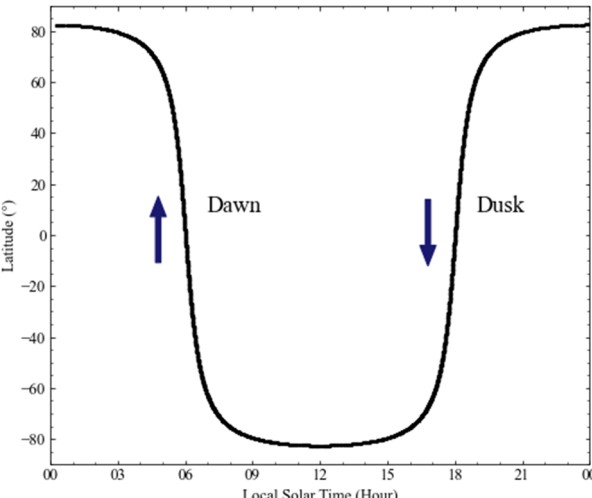

**Figure 2.** Local solar time versus latitude coverage of the QQ-Satellite orbit.

### 2.2. Processing Strategy

The results in this research were created by considering the nongravitational empirical acceleration from the POD processing procedure and the drag acceleration from the atmospheric model to estimate the density. The original atmospheric drag acceleration in orbit dynamics is computed as follows [27]. Due to the uncertain parameters, such as the inaccurate computation of the drag coefficient and the frontal region area, in Equation (1), the limited ability to accurately calculate the atmospheric drag perturbation acceleration $a_{drag}$ is a challenging task:

$$a_{drag} = -\frac{1}{2}\frac{C_D A}{m}\rho v_r^2 \vec{e}_v \tag{1}$$

As expected, $C_D$ is the dimensionless atmospheric drag coefficient, m is the satellite mass, and *A* is the frontal region in the orientation of the satellite motion. If the satellite inertial velocity is $v$ and the atmosphere at that point is $v_{atm}$, then the satellite velocities relative to the atmosphere are $v_r$, $v_r = v - v_{atm}$. Here, $\rho$ is the instantaneous thermospheric mass density. The unit vector $\vec{e}_v$ of the satellite velocity vector $\vec{v}$ neglects the influence of atmospheric movement. All the position and velocity values in the state vectors are included in the framework of the Earth-Centered Inertial (ECI) coordinate system.

The majority of the POD solutions of the acceleration of LEO satellites are assessed by concurrently evaluating the satellite drag coefficient. Here, the estimated acceleration absorbs the bias in the assumed fixed area-to-mass ratio and the average error in the empirical atmospheric mass density model over the orbital arc. However, the diameter of the QQ-Satellite of the spherical shell is 800 mm. Due to the compact spherical shape, and the homogeneous spherical structure, the cross-sectional area can be computed accurately at any attitude and set as a fixed value of 2.011 m$^2$. Then, the area-to-mass ratio can be set as a fixed quantity.

With respect to a spherical satellite, its precise orbital data can be determined to a resolution of a few centimeters with high-precision GNSS receivers from its obtained dense BDS and GPS tracking observations. The GPOD receiver is equipped with the QQ-Satellite as the high-precision GNSS receiver, which was designed and developed by the NSSC. We employed the reduced-dynamic POD as the routine positioning mode, as well as during periods of geomagnetic disturbance. The Radial–Transverse–Normal (RTN) reference coordinate frame was used to evaluate the accuracy of the POD. In fact, as demonstrated in Figure 3, the 10-day root mean square (RMS) errors of the POD in the geomagnetic quiet period, corresponding to the overlap error, were analyzed. The accuracy of the QQ-Satellite POD measurements in the along-track (T) direction is considered to be better than 2 cm, and the accuracy of the three-dimensional (3D) RMS is better than 3 cm. As a result, the precise state vector and instantaneous Kepler elements of the QQ-Satellite can be calculated by interpolation at any time period. In our research, we set the precise orbit ephemeride time step to 30 s.

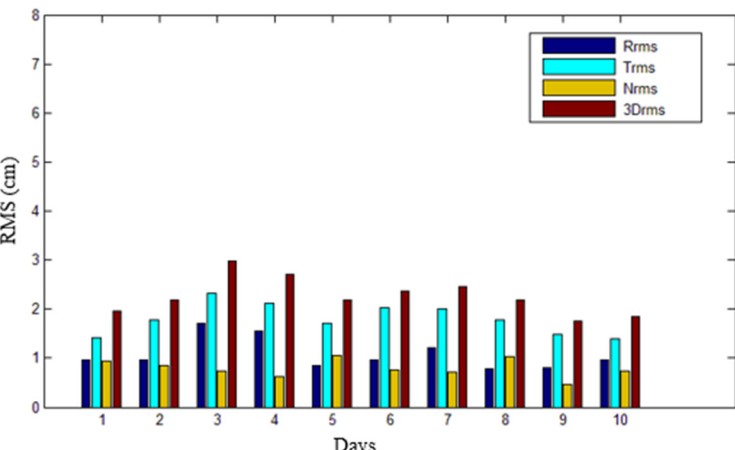

**Figure 3.** QQ-Satellite statistics of the accurate analysis of the precise orbit determination. Rrms, Trms and Nrms indicate the root mean squared (RMS) errors in the radial–transverse–normal (RTN) directions, respectively.

Determining the drag coefficient of a satellite is a much more difficult problem because it is associated with gas surface modeling. Even though several gas surface models have been developed and investigated for their consistency, the confirmed effectiveness of these models is very inadequate [28]. Given that the QQ-Satellite has a compact spherical shape, and its drag coefficient is fundamentally unrelated to the direction of orbital along-track motion, a drag coefficient model can be fixed to investigate variations in the atmospheric density. In addition, the QQ-Satellite is in a near-circle orbit, the surface materials are well

distributed, and the calculated drag coefficient based on Cook's formulation is basically the same. Thus, the resulting $C_D$ for average environmental conditions is a calculated fixed value of 2.205 globally [29]. This value is used to derive the mass density from the QQ-Satellite POD process and the orbit propagation. Hence, we focused on estimating and validating the densities. This fixed drag coefficient will introduce the same systematic bias in both the derivation and validation. To achieve the ideal trade-off between the accuracy and the resolution in various tracking situations, additional detailed investigations are needed.

Then, the atmospheric mass density can be estimated once the acceleration attributable to atmospheric drag is known. Similar to the developed method [18], which was chosen to estimate a stochastic acceleration in the direction of drag, a properly selected set of estimated stochastic accelerations can assume the role of the errors in the atmospheric drag. A constant drag coefficient is also employed. Moreover, empirical deterministic acceleration functions were used in this study. The drag perturbation acceleration includes both the atmospheric density model drag acceleration and the empirical force acceleration of the POD in the velocity direction.

In reduced-dynamic POD solutions, estimating the empirical acceleration not only compensates for the bias force in the background models, but also reduces the effect of errors that caused by the force model. For the near-circular QQ-Satellite orbit, the empirical acceleration is estimated from the orientation of the along-track direction. Thus, at each updated interval, this estimation absorbed all the bias and errors of the theoretical drag force model.

The approximate value of the total atmospheric density $\rho$ can be estimated based on the following equation:

$$acc_D = -\frac{1}{2}\frac{C_D A}{m}\rho_m v^2 \tag{2}$$

$$\rho = (acc_D + acc_E)/\left(-\frac{1}{2}\frac{C_D A}{m}v^2\right) \tag{3}$$

On the right side of Equation (3), all the acceleration terms can be obtained from reduced-dynamic POD solutions. Because the reduced-dynamic POD depends solely on the force model that governs the satellite movement and tracking observations, the force model errors are not compensated. Consequently, this strategy is based on the batch least square reduced-dynamic POD method. This process is performed in the GNSS high-precision orbit determination procedure that is used to transform the carrier phase and code observations of the QQ-Satellite obtained from both BDS and GPS into nongravitational acceleration. Therefore, at each update time, the reduced-dynamic POD procedure yields accurate estimates of the empirical drag acceleration $acc_E$ in the along-track direction.

When determining the orbit of LEO satellites, an empirical atmospheric density model is commonly used to calculate the drag acceleration; thus, a background density model ($\rho_m$) is utilized to calculate the nominal drag acceleration $acc_D$. Figure 4 shows the processing flow chart for retrieving the atmospheric density from precise orbit ephemerides.

*2.3. Orbit Propagation*

This section introduces a validation application of QQ-Satellite-derived densities. An available empirical atmospheric density model employs the QQ-Satellite GNSS-derived precise orbit solution observations to be calibrated. Then, the calibrated density model is applied in the initial orbit propagation. This calibration approach is based on the basic theory of using the derived densities to redress the density parameters involved in a baseline model. In this study, the developed NRLMSISE-00 density model was used as the baseline atmospheric model. This model has the advantage of availability and low computational complexity; a wide range of applications also allows the result to be more generalizable. Then, the calibration input served as the POD-derived density over a period of 24 h from the start time. Finally, the orbits of the QQ-Satellite were propagated using the calibrated model, also beginning from the start of the calibration time period.

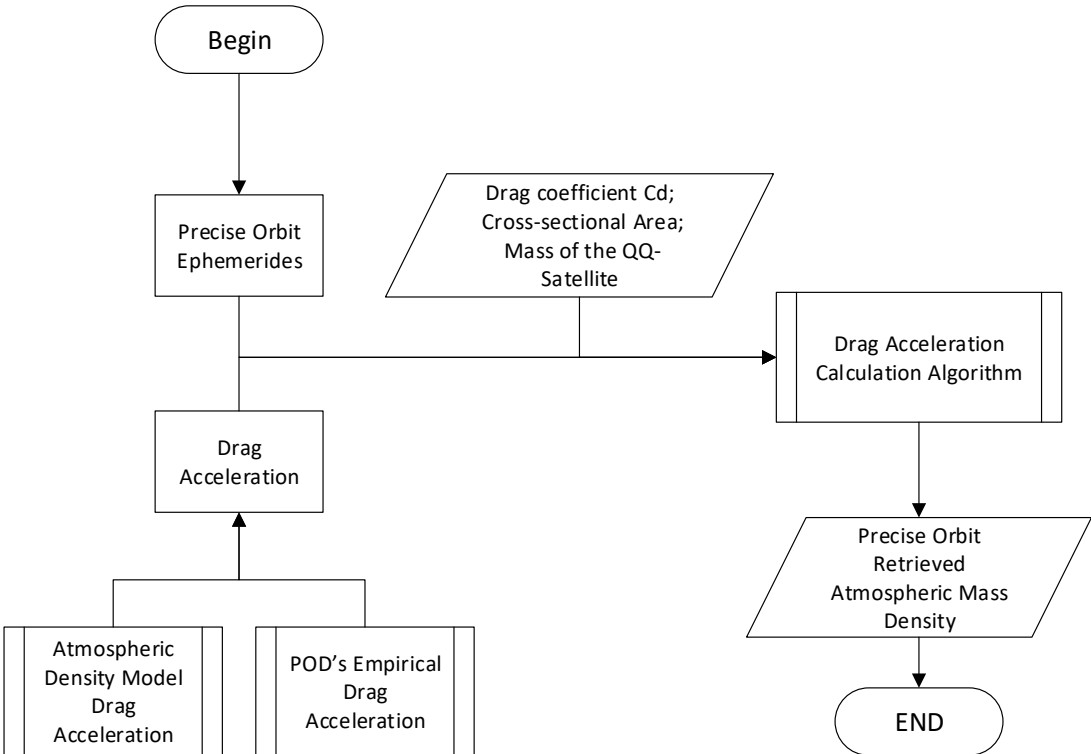

**Figure 4.** The method for retrieving densities from precise orbits.

The configurations of the orbital dynamics used in this study are listed as follows:

- Geopotential using the EGM-96 [30] to the degree and order of $120 \times 120$;
- CSR 3.0 [31] for the ocean tidal model truncated to $30 \times 30$;
- Solid earth and pole tides using the IERS2010 conventions [32];
- DE 430 planetary ephemeris [33] for third-body perturbations by the sun and moon;
- Radiation pressures, mainly including the solar and Earth radiation pressures; and
- Atmospheric drag, considering a corotated atmosphere for calculating the horizontal wind velocity. The spherical QQ-Satellite cross-sectional area is 2.011 m$^2$.

Modeling the geopotential field using the spherical harmonics truncated at $120 \times 120$ could provide satisfactory accuracy for orbit propagation. We did not consider the higher-order solid Earth and ocean tides and gravitational terms, but these factors can be deliberated on in future research. A reflectivity coefficient of 1.0 was used to model the solar radiation pressure. A fixed drag coefficient $C_D$ of 2.205 was used as described above. The correspondence between the predicted trajectory and precise orbit ephemeris is considered to be acceptable once the $C_D$ and atmospheric densities are properly evaluated. In other words, without knowing an accurate $C_D$, the atmospheric density in orbit propagation applications cannot be well estimated because the two factors are highly correlated.

The orbits of the QQ-Satellite were propagated in the Earth-Centered Inertial (ECI) co-ordinate system of the J2000 reference framework using the Cartesian position and velocity, whereas the Earth-fixed International Terrestrial Reference System and Frame (ITRF)-2010 reference frame was employed in the process of calculating the drag and geopotential accelerations. The ITRF-2010 and J2000 reference frames and leap-seconds kernel from NASA's SPICE toolbox [34] were utilized both for the sun and moon ephemerides and the time transformations and reference frames.

The orbit propagation results were finally compared in the RTN reference coordinate frame. Here, radial (R) means the direction parallel to the extension of the geocentric radius that intersects the mass center of the satellite. The along-track (T) direction is orthogonal to the radial direction and the state vector defined satellite plane. Together, the

R, T, and cross-track (N) directions are determined by forming a right-handed orthogonal system [35].

## 3. Results

### 3.1. Atmospheric Mass Density Derived from the QQ-Satellite POD Observation

The density estimation mainly uses the equation of perturbation due to drag. In this study, the densities were estimated during the geomagnetic activity maximum. The correlation results show the estimated mass densities, which were determined by deriving the drag acceleration with the aforementioned processing strategy. Note that the total drag accelerations are available as Level 2 products for the QQ-Satellite mission. The derived densities are delivered every 30 s.

Figure 5 shows an instance of the evaluated densities and model densities in ascending and descending orbit from the POD data on 1 November 2021. In Figures 5 and 6, from the descending orbit (dusk), the atmospheric densities in the Southern Hemisphere in summer are greater than those in the Northern Hemisphere in winter on a low-geomagnetic activity day, but this phenomenon is not clear in the ascending orbit (dawn).

At the same quiet geomagnetic time, atmospheric densities at dusk (near 18:00 local time) are greater than those at dawn (near 06:00 local time). These findings indicate the characteristics of the local-time variation in the thermospheric mass density, as also depicted in Figure 6 for the two selected laps of the QQ-Satellite orbit selected. High temperatures in the summer afternoon cause the air to heat up and lift from the surface of Earth to the upper atmosphere. Moreover, the midnight mass density maximum has been observed from the QQ-Satellite. This feature is a night-time enhancement in the thermosphere and is also associated with the temperature increase to a maximum at midnight.

The NRLMSISE-00 model in Figures 5 and 6 also demonstrates that the density exhibits a slight increase at midnight. Although it is not apparent that the atmospheric density in the Southern Hemisphere in summer is greater than that in the Northern Hemisphere, evidence indicates that the atmospheric density at dusk is greater than that at dawn. In addition, the density values are greater in the equatorial area than in the northern and southern hemispheres at dusk, and the performance of physical mechanisms is closer to the average at both dawn and dusk.

Figures 7 and 8 show an example of atmospheric density enhancement due to magnetic storm forcing. For comparison, a color bar with the same magnitude is used, but the true magnitude of the geomagnetic activity disturbance density can be extremely large. While analyzing the atmospheric physical mechanisms (Figure 7), the QQ-Satellite derived densities reveal that during magnetic storms, ring current energy is injected into the South Pole. This energy heats the atmosphere, drives the atmospheric density uplift at the South Pole, and then spreads to lower latitudes. In contrast, the NRLMSISE-00 model considers a geomagnetic storm as simultaneous heating and uplift at both poles. In addition, the time delay response is significantly different, and the POD-derived densities are correspondingly earlier than those of the model.

The analysis of the global atmospheric density in single-lap orbits can be easily seen in Figure 8, which includes the (a) beginning period, (b) main phase of the geomagnetic storm, and (c,d) recovery phase. Density values begin to spread from the Antarctic to lower latitudes, which validates the previous analysis. The maximum density is achieved near the equator.

Then, we analyzed the variation in the atmospheric density for a full-time period. In the early phase of the launch of the QQ-Satellite mission, the highest densities were ascribable to the comparatively extreme geomagnetic activity caused by coronal mass ejections in this time period. For comparison, Figure 9 illustrates the time series densities of the QQ-Satellite POD-derived and NRLMSISE-00 atmospheric models from the beginning of November until the 8 November 2021. Using the current processing strategy, high-accuracy densities were obtained from the indirect evaluation of the total drag accelerations in the POD, while high-fidelity QQ-Satellite aerodynamic and geometry models were employed.

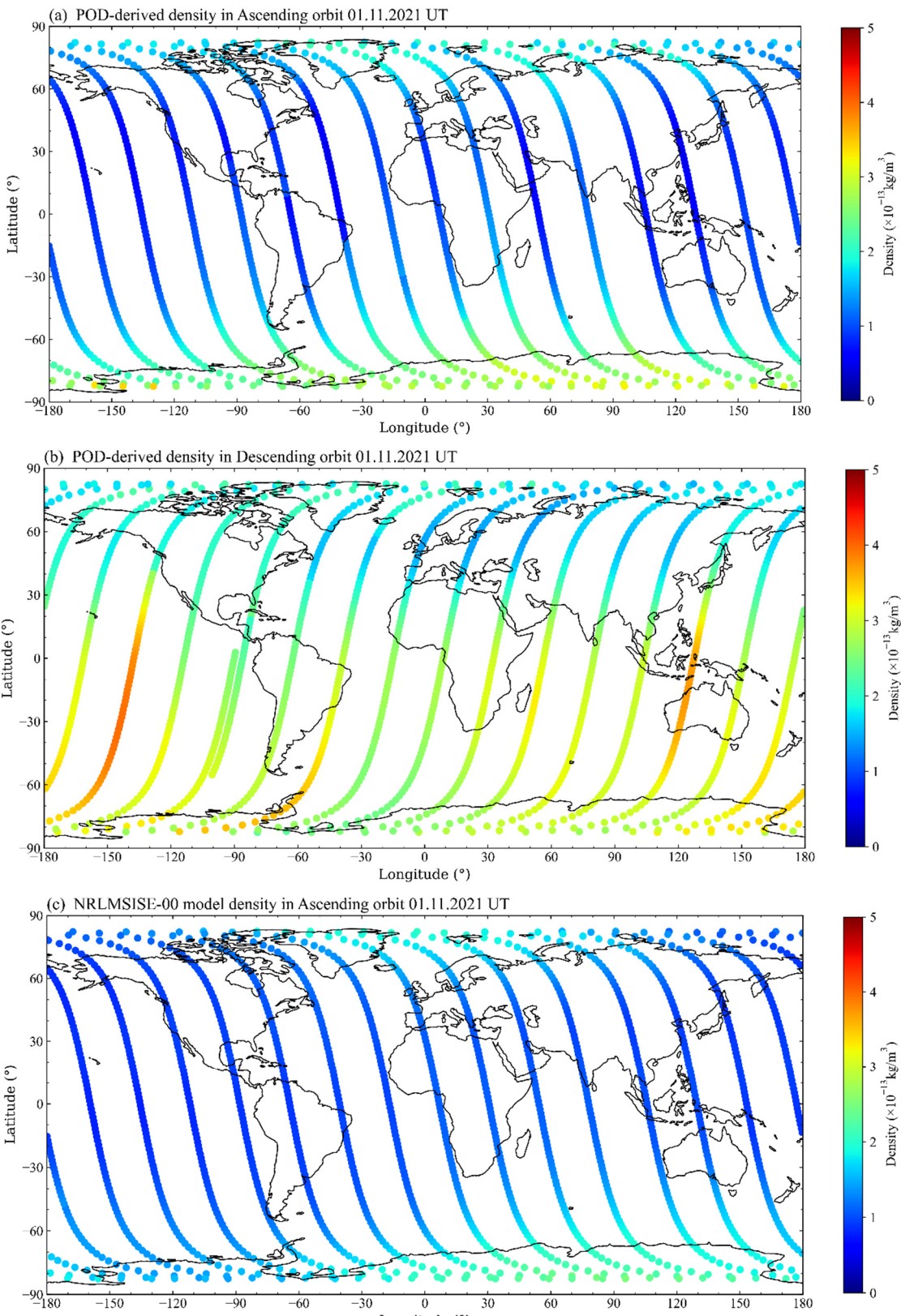

**Figure 5.** *Cont.*

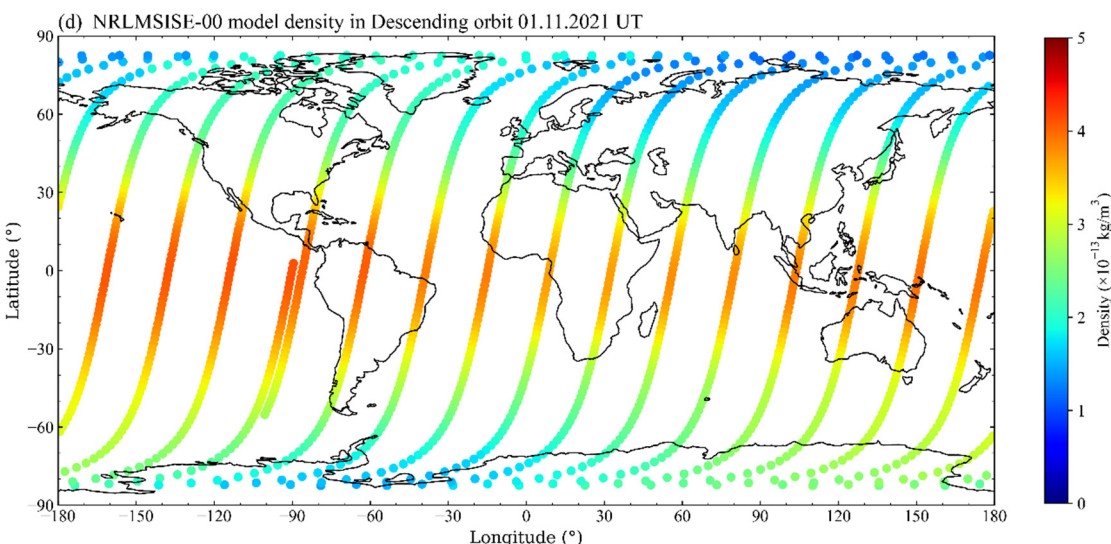

**Figure 5.** QQ-Satellite-derived atmospheric density in the (**a**) ascending and (**b**) descending orbits; NRLMSISE-00 model density in the (**c**) ascending and (**d**) descending orbits on 1 November 2021 UTC time.

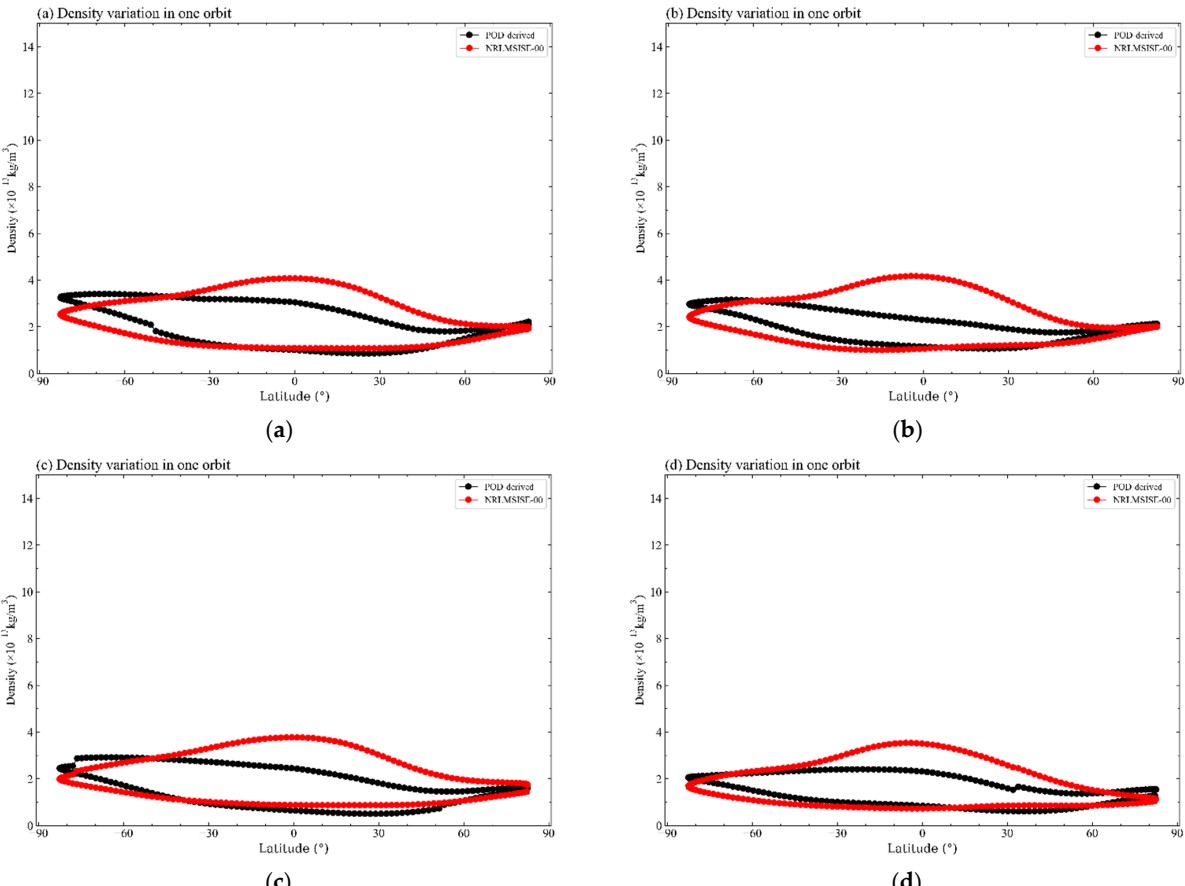

**Figure 6.** The density variations during one orbital period, especially in the quiet period. (**a**,**b**) Description of the density variations on 1 November 2021. (**c**,**d**) Description of the density variations on 2 November 2021. The image order reflects the chronological order of the exact day. The geomagnetic activity was moderate on these days. The upper part of the orbital arc is the descending orbit at dusk (near 18:00 local time), and the lower part of the orbital arc is the ascending orbit at dawn (near 06:00 local time).

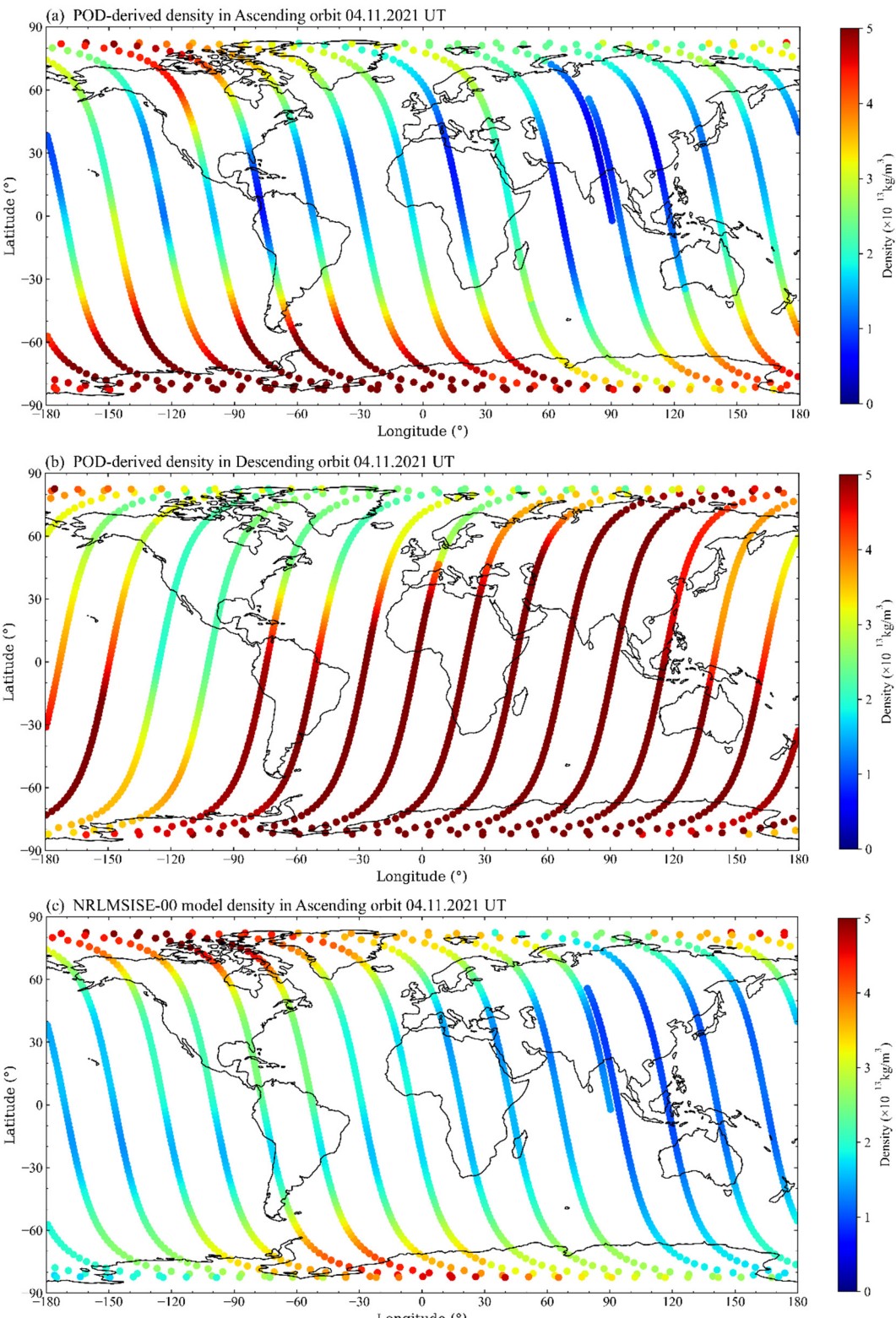

**Figure 7.** *Cont.*

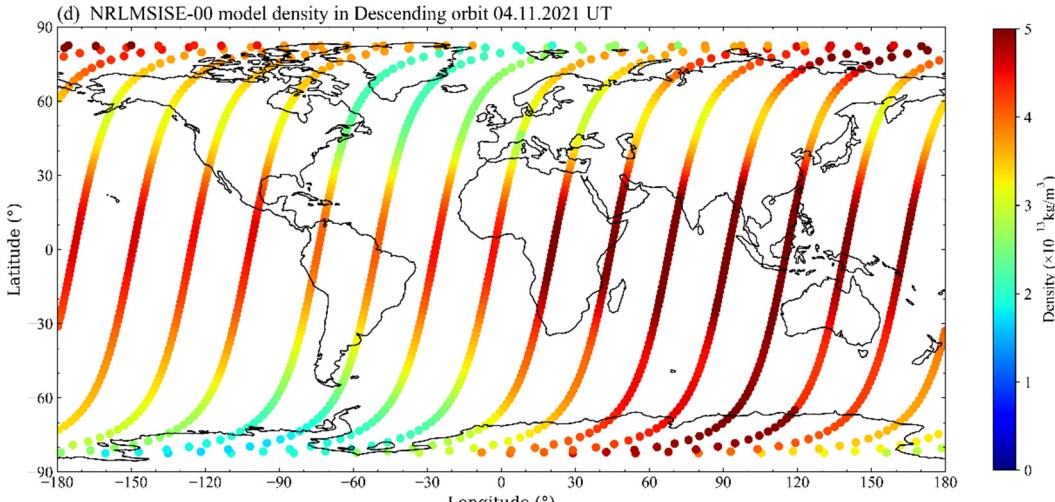

**Figure 7.** QQ-Satellite-derived atmospheric density in the (**a**) ascending and (**b**) descending orbits; NRLMSISE-00 model density in the (**c**) ascending orbit and (**d**) descending orbits on 4 November 2021 UTC time.

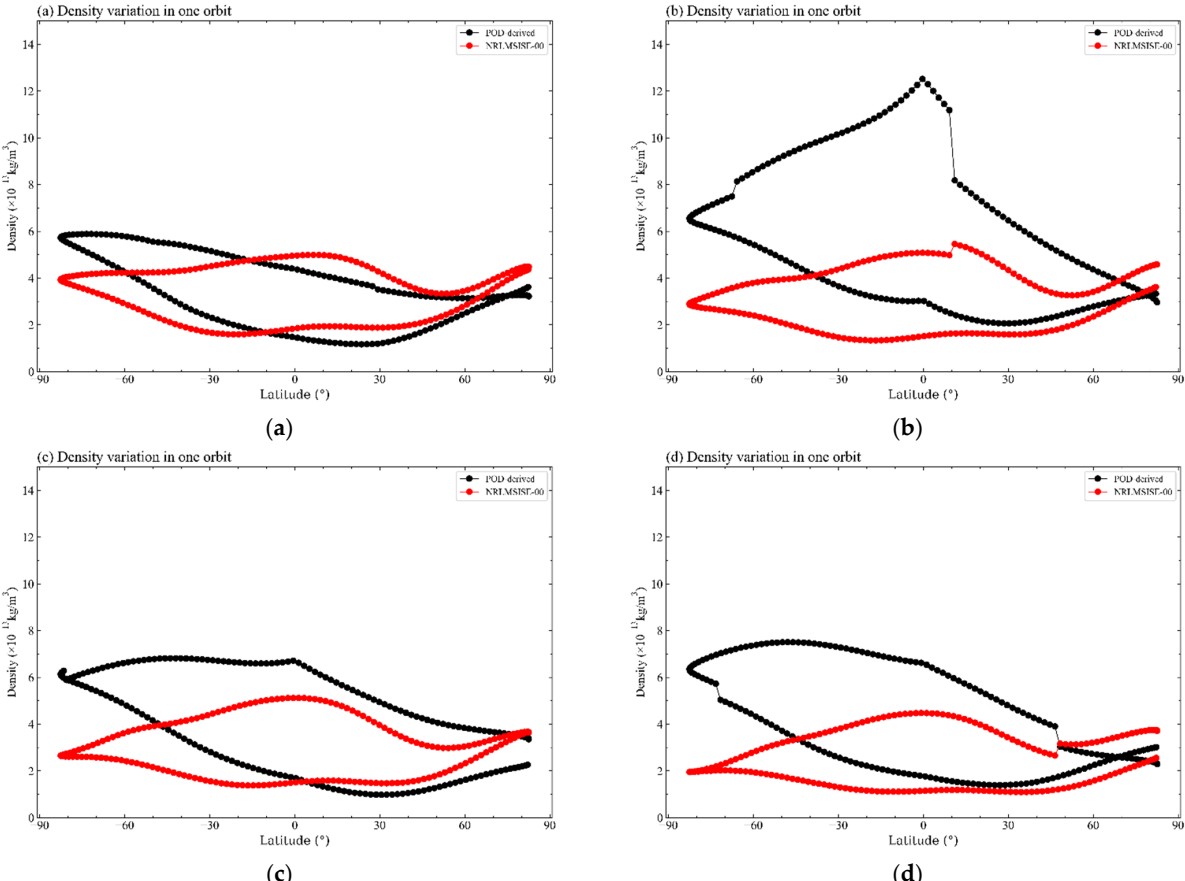

**Figure 8.** The density variation during one orbital period. (**a**–**d**) Description of the density variations on 4 November 2021. The image order is the chronological order of the exact day. (**a**) The orbital density before the geomagnetic storm. (**b**) The main phase of the QQ-Satellite experiencing a geomagnetic storm of a lap of orbit. (**c**,**d**) The recovery phases. The upper part of the orbital arc is the descending orbit at dusk (near 18:00 local time), and the lower part of the orbital arc is the ascending orbit at dawn (near 06:00 local time).

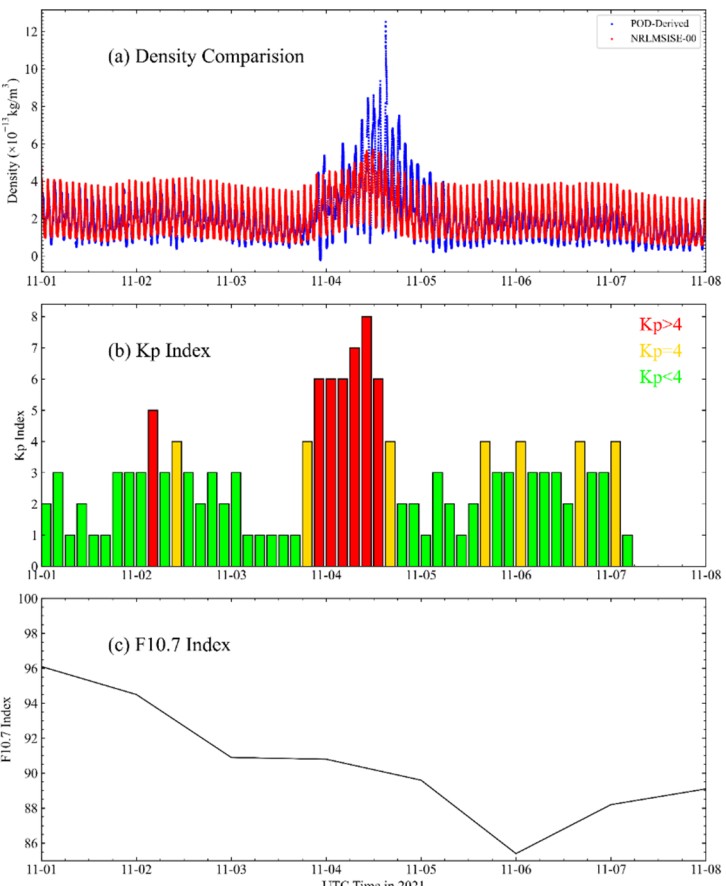

**Figure 9.** (**a**) A temporal comparison between the POD-derived densities and the NRLMSISE-00 atmospheric density model results; (**b**) Kp index; and (**c**) F10.7 index. The first epoch is noted at 21 h on 1 November 2021 (UTC).

As a reference, space weather indices (also see Appendix A) were introduced to explain the external disturbance. The middle part of Figure 9 shows the geomagnetic index Kp, and the bottom part of Figure 5 indicates the solar activity proxy $F_{10.7}$. These indices were intended to better understand the trajectory modeling of the LEO satellite. The overall increase in the POD-derived densities is consistent with the sudden increase in Kp and the gradual decrease in $F_{10.7}$. At the end of the last day, the blank area in the Kp index figure was Kp = 0.

In the initial geomagnetic minimum conditions, the QQ-Satellite experiences average derived densities of less than $2.45 \times 10^{-13} \mathrm{kg/m^3}$. However, the maximum estimated density was greater than $12.53 \times 10^{-13} \mathrm{kg/m^3}$, which is clearly noticeable as the peak of the blue line in Figure 9. However, the NRLMSISE-00 density model overestimated the densities during the geomagnetic minimum, in which the average value was more than $2.91 \times 10^{-13} \mathrm{kg/m^3}$ during the quiet period. In the geomagnetic maximum, the peak density value was only $5.89 \times 10^{-13} \mathrm{kg/m^3}$. An approximately 18.77% difference is noted in the POD-derived density. However, during the geomagnetic storm, the one standard deviation difference was on the magnitude of 52.99%. This result suggests that the POD-derived atmospheric densities were more sensitive to geomagnetic storms, which is consistent with the results obtained from the solar and geomagnetic activity indices.

For the time period in November 2021, the Pearson correlation coefficient was used to estimate the correlation between the POD-derived densities and the NRLMSISE-00 densities, as shown in Table 1. The Pearson correlation method is the most commonly used method for numerical variables. This method assigns a value between −1 and 1, with 1 indicating a totally positive correlation, 0 indicating no correlation and −1 indicating a totally negative correlation [36]. The table clearly indicates that approximately all Pearson

coefficients are greater than 0.7. These results suggest that the derived density and the model results exhibit a significant and positive relationship. Even during the extreme geomagnetic storm, the Pearson coefficient was 0.686, and a distinct positive relationship was still observed.

**Table 1.** Comparison between the POD-derived densities and NRLMSISE-00 densities.

| UTC Time in 2021 | Correlation Coefficient | Mean Difference |
| --- | --- | --- |
| 1 November | 0.823 | −3.41% |
| 2 November | 0.762 | −4.21% |
| 3 November | 0.725 | −8.74% |
| 4 November | 0.686 | −27.1% |
| 5 November | 0.749 | −6.22% |
| 6 November | 0.758 | −8.14% |
| 7 November | 0.833 | −10.8% |

The mean difference between the POD-derived density and the model density was computed by $\left[\sum_{i=1}^{N}(\rho_{E,i} - \rho_{M,i}/\rho_{M,i})\right]/N$, where $N$ is the density points in one day, and $\rho_{E,i}$ and $\rho_{M,i}$ are the estimated density and the model density, respectively, at epoch $i$. The mean difference calculation values are shown in Table 1. The average mean difference between the derived density and the NRLMSISE-00 model density is less than 11%. The mean difference increased to 27.1% on the day of the geomagnetic perturbation, which is almost three-fold greater than that noted in the beginning and recovery phases.

In the first two parts of Figure 10, density variations caused by solar and geomagnetic activities can be observed followed by diurnal and latitudinal variations, which can be clearly identified. As expected, for all latitudes, the density variation from the quiet phase to the storm phase is positive. This finding also verifies that the densities increase significantly from the quiet phase to the geomagnetic storm maximum. Figure 10 shows the response of the atmospheric densities to the extreme geomagnetic storms, the coverage of which is demonstrated in time series figures: (a) the POD-derived density; (b) the NRLMSISE-00 model density, where the density observations are supported in epoch time intervals × latitude bins with a volume of 3 h × 2°, and the mean density values of the grid are shown in the color bar; and (c) the geomagnetic index Dst (also see Appendix A). The time series of the starting epoch is considered to be the beginning of the main magnetic storm phase. The Dst index rather than Kp was used because Dst describes the ring current energy stored in the atmosphere and has a higher temporal resolution during the geomagnetic storm main phase, making it more precise than the Kp index.

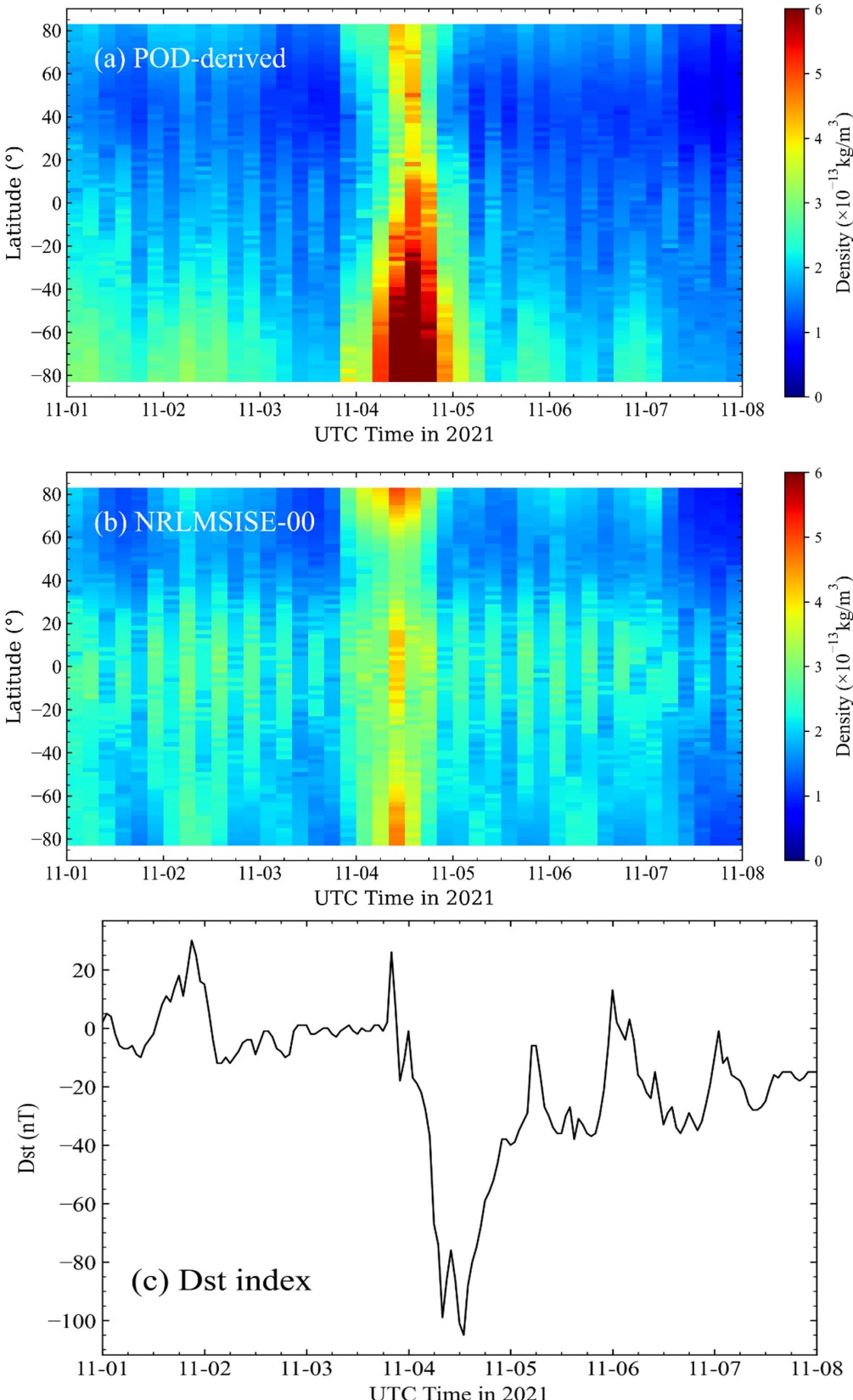

**Figure 10.** A spatial comparison between (**a**) the derived densities and (**b**) the NRLMSISE-00 atmospheric density model results; (**c**) the Dst index. The first epoch is noted at 00:00 h on 1 November 2021 (UTC).

From these figures, the relationship between the two groups of densities indicates a clear temporal match. On 4 November, the Dst index decreased rapidly, from −5.4 nT (weak geomagnetic activity $|D\text{st}| \leq 50$ nT) to −107.3 nT (intense geomagnetic activity $100$ nT $\leq |D\text{st}|$). These results indicate that geomagnetic storms occurred in the Southern Hemisphere (Antarctic region) and suddenly appeared in a very short time period (sudden storm commencement). The densities retrieved from the QQ-Satellite POD data are highly consistent with the NRLMSISE-00 atmospheric model and Dst index.

For the initial magnetic minimum condition, the QQ-Satellite experienced atmospheric densities under the model, which are visible as the deep blue area and clearly found near the equator. Under the interval of the higher geomagnetic levels, the derived density is significantly higher than the model calculation result, suggesting that the NRLMSISE-00 atmospheric model underestimates the density during geomagnetic activities. The NRLMSISE-00 model exhibits an obvious symmetry around the equator.

After the end of the magnetic storm period, the model density is reduced more quickly and then resumes the average interval. However, the derived density remained responsive until midnight on 4 November 2021. The ring current energy continually decreased in the period of the storm recovery phase. The Dst index maintained its increase until the end of the geomagnetic storm, and then the perturbation of the geomagnetic field ceased.

Finally, several days of precise orbit information were handled, and the effectiveness of the approach that derives the thermospheric densities from the total nongravitational accelerations was proven in this study. We plan to develop a validation method to verify the derived atmospheric density improvement on the accuracy of orbital propagation. The orbit propagation is performed in a numerical propagator. Additional experiments will be designed to process the precise orbit data of other satellite missions, including orbits with extensive vertical (higher and lower altitudes) coverage.

From our preliminary research, the atmospheric densities derived from the empirical accelerations of the QQ-Satellite contain systematic bias and uncertainties, and corresponding validation methods will be performed in subsequent experiments. We intended to analyze the POD-derived atmospheric mass density in this study, and the mass spectrometer measurements will be added as comparison data for integrated analysis in future research.

### 3.2. Application of the QQ-Satellite Derived Mass Density for Improved Orbit Prediction

Improving the accuracy of short-term orbit propagation is the main purpose of estimating atmospheric densities. The position vector of the QQ-Satellite precise orbit data was applied as the reference orbit to evaluate the derivations of orbit propagation. We compared the propagation errors, which are the atmospheric densities obtained from the original NRLMSISE-00 density model densities and the QQ-Satellite derived densities. Figure 11 exhibits an example of the QQ-Satellite orbit propagation bias in the RTN local orbital reference frame, which was transformed from the J2000 reference frame, without and with the calibration of the NRLMSISE-00 model using the derived densities. Because the QQ-Satellite is in a near-circle orbit, the T direction in the RTN coordinate frame is aligned to the along-track direction, and large along-track biases are generally caused by large density model errors. The main comparison was between the along-track biases. Compared with the along-track bias, the deviations of the radial and cross-track are almost negligible. During low geomagnetic activities, the orbit was propagated for 24 h (since 1 November 2021). Using the original NRLMSISE-00 atmospheric model in the along-track position, the residual was 110.82 m. However, these errors were downscaled to approximately 90.36 m when the derived densities were employed in the calibration, for an improvement of approximately 18.45%.

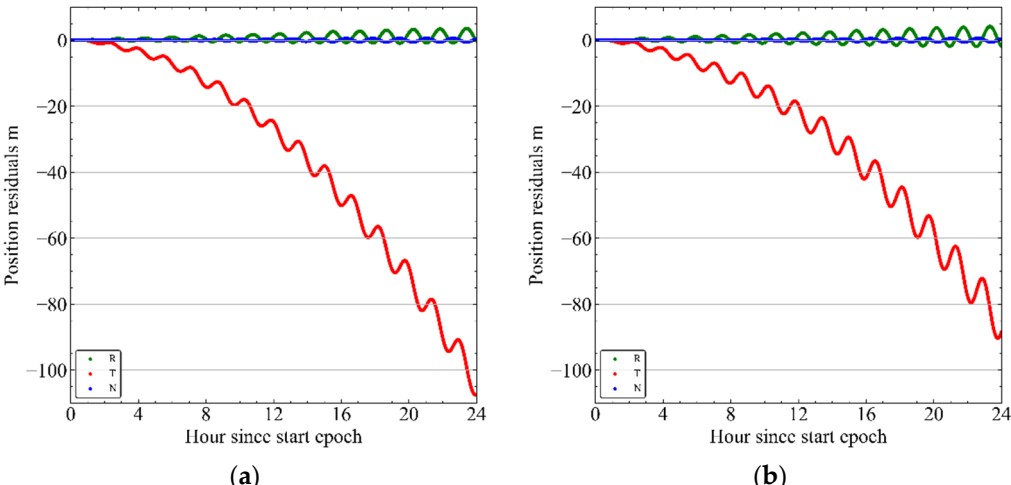

**Figure 11.** At low magnetic activities, orbit propagation error magnitudes, without (**a**) and with (**b**) calibration of the derived densities. R is the radial direction, T is the along-track direction, and N is the cross-track direction.

Atmospheric densities derived from nongravitational acceleration data of precisely tracked LEO spacecraft, such as the QQ-Satellite, could play a significant role in the calibration and development of accurate empirical thermospheric mass density models. A simulation study [37] demonstrated that an orbit propagation accuracy of 100–200 m is achievable in 24 h, and the basic idea is to employ the model coefficients for modification. However, our new method of deriving density from the drag perturbation acceleration of POD observations is better than 100 m for 24 h.

Then, propagation was applied on a high geomagnetic activity day to further demonstrate the effectiveness of the POD-derived density in improving the accuracy of orbit propagation. Figure 12 is a comparison of the propagated biases along-track for 24 h at midnight on 4 November 2021. For the bias using the original NRLMSISE-00 density model, its magnitude is boosted almost monotonically to greater than 405.51 m. Given that the models are not sensitive to space weather activities, this property is common to almost all empirical density models under these conditions. However, after using the QQ-Satellite-derived densities, the along-track bias is reduced to 172.48 m at the end of the prediction time, representing a significant improvement of 57.4% in the propagation accuracy.

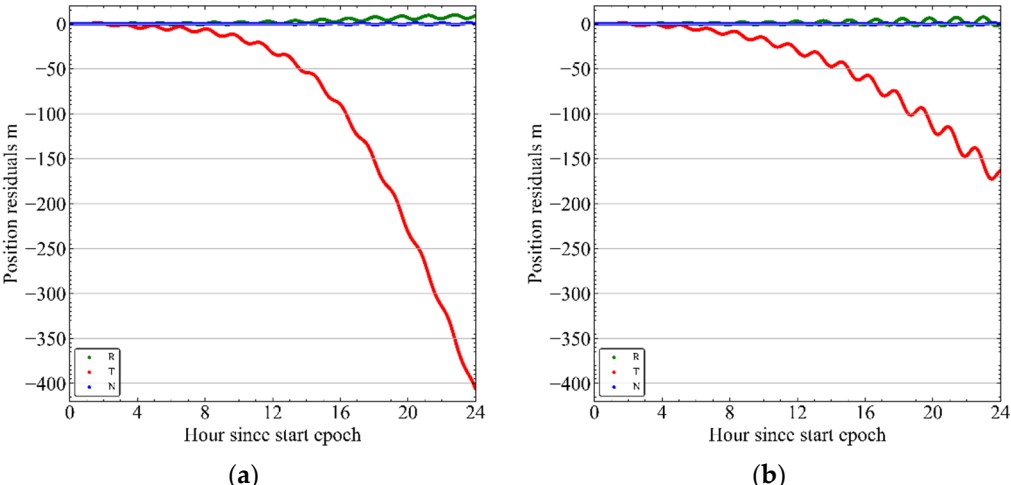

**Figure 12.** At high magnetic activities, orbit propagation error magnitudes without (**a**) and with (**b**) calibration of the derived densities. R is the radial direction, T is the along-track direction, and N is the cross-track direction.

The root-mean-square values of the 3D trajectory differences between the propagated orbits and the precise orbit ephemeris are shown in Table 2. This result clearly indicates that the 3D orbit derivations during periods of extreme geomagnetic activities are considerably greater than those noted during periods of quiet geomagnetic activities. The 3D RMS position errors improve slightly by 11.51% in periods of low geomagnetic activity. During periods of higher geomagnetic activity, with the decrease in the peak 3D error from 85.51 m for the NRLMSISE-00 atmospheric model to 40.89 m for the QQ-Satellite-derived densities, the reduction in the propagation error was greatly increased by 52.18%. The 3D RMS improvement percentage in different situations was approximately 40.67%. Therefore, the error reduction is significant.

**Table 2.** Comparison of the derived density and model density of 24 h orbit propagation of the 3DRMS without and with geomagnetic storms.

| UTC Time | 3D RMS Model | 3D RMS Calibrated | Improved Percentage |
|---|---|---|---|
| 2021.11.1 | 25.46 m | 22.53 m | 11.51% |
| 2021.11.4 | 85.51 m | 40.89 m | 52.18% |

In summary, the above preliminary research has revealed that the derivation of thermospheric mass densities from POD data is not only practical but also reasonable and effective in improving the accuracy of orbit propagation.

## 4. Discussion

The QQ-Satellite, which was designed to research the response mechanism and characteristics of the atmospheric environment under the disturbance of solar and geomagnetic activity, is China's latest spherical satellite launch. To achieve the aforementioned scientific goals, the QQ-Satellite is equipped with an onboard high-precision GNSS receiver GPOD. This receiver tracks both GPS and BDS positioning observation data and employs a reduced-dynamic POD procedure to produce high-accuracy precise orbit data and proper drag perturbation acceleration data. At present, most satellites are equipped with onboard GNSS navigation receivers, so this research is conducive to deepening the understanding of the orbital atmospheric mass density from the precise orbit of GNSS observations and has great value and potential.

In addition, a solution is proposed for determining the thermospheric mass density from nongravitational accelerations of the QQ-Satellite, and more investigations are intended to process precise orbit data from other satellite missions. The fundamental perturbation equation due to atmospheric drag is the basic idea of this approach; drag acceleration includes both the atmospheric density model drag acceleration and the empirical force acceleration of POD in the velocity direction. The along-track empirical acceleration is efficiently determined by reduced-dynamic POD and has an excellent temporal resolution and precision. In addition, the method aims to increase the accuracy of orbit propagation. The derivation algorithm and strategy can be straightforwardly implemented within the framework of any POD procedure with an onboard GNSS receiver satellite. This concept was validated using the NRLMSISE-00 atmospheric model and the QQ-Satellite GNSS positioning data.

For the QQ-Satellite, the higher spatial and temporal resolution and accurate densities derived from the POD procedure improve orbit propagation and provide insights into variations in the density of the upper atmosphere density. The overall POD-derived density exhibits good consistency with the solar index $F_{10.7}$ and the geomagnetic indices Kp and Dst. In particular, the QQ-Satellite POD-derived densities better estimate the actual variations in atmospheric density and more sensitively and accurately model the density during geomagnetic activity maxima. Moreover, given that the atmosphere is strongly forced by solar and geomagnetic activity, the above enhancement improves the modeling of the relationship between the thermospheric density and the drag acceleration.

The application of the derived density from drag accelerations to reduce orbit propagation errors was presented to demonstrate the effectiveness of the method. When the original NRLMSISE-00 density model is calibrated using the QQ-Satellite-derived density, both without and with higher geomagnetic activity, the maximum orbit propagation residuals over a 24 h period were reduced from 110.82 m to 90.36 m and from 405.51 m to 172.48 m, respectively. The error reduction during geomagnetic activity is significant.

In addition, new model development and density model calibration urgently require more density knowledge over a wider space, including a large vertical height and longer temporal periods, and retrieving upper atmospheric mass density from precise orbit measurements of LEO satellites, which are well suited to meet these requirements. More short-term spacecraft and an increased accuracy in debris orbit propagation are crucially significant to space situational awareness. In future work, we plan to further enhance the gas surface modeling and develop a more realistic radiation pressure model of the QQ-Satellite. Because the NRLMSISE-00 model is relatively less sensitive to space weather activity, we also plan to further consider physics-based models, such as the Thermosphere Ionosphere Electrodynamics General Circulation Model (TIE-GCM) [38], Coupled Thermosphere Ionosphere Plasmasphere Electrodynamics Model (CTIPe) [39], and the Global Ionosphere–Thermosphere Model (GITM) [40]. These models can compete with empirical models during quiet phases and have significant potential to forecast the atmospheric variation caused by geomagnetic disturbances. In addition, mass spectrometer measurements will be added to analyze the estimated density accuracy and provide information to improve existing models for future efforts.

## 5. Conclusions

The QQ-Satellite equipped with a high-precision GNSS receiver can track both GPS and BDS signals, and subsequent routine POD procedures can provide empirical accelerations in the along-track direction with sufficient accuracy. Empirical acceleration from POD processing and the drag accelerations calculated by the atmospheric density model are feasible and valid for retrieving atmospheric mass densities.

The QQ-Satellite-derived densities elaborate the intra-diurnal spatiotemporal (latitudinal and longitudinal) variations, which also provide insights for understanding variations in the density of the upper atmosphere. Data from several days of retrieved densities were used to enhance the accuracy of orbit propagation, and the results are promising, especially under high-geomagnetism conditions.

**Author Contributions:** Conceptualization, Y.S. and B.W.; methodology, X.M. and B.W.; software, X.M. and B.W.; validation, B.W., X.T. and F.Y.; formal analysis, X.Z., B.G., S.W. and H.Q.; investigation, W.B., Y.L. and X.W. (Xinyue Wang); resources, Y.S. and P.H.; data curation, X.M. and X.W. (Xianyi Wang); writing—original draft preparation, Y.S. and B.W.; writing—review and editing, B.W., Y.S. and X.M.; visualization, Q.D.; supervision, X.M.; project administration, Y.C. All authors have read and agreed to the published version of the manuscript.

**Funding:** This research was funded by the National Natural Science Foundation of China, grant numbers 42074042 and 41775034.

**Data Availability Statement:** QQ-Satellite (ephemeris) datasets can be obtained from the repository provided by the National Space Science Center. The NRLMSISE-00 empirical model source code is available at ftp://hanna.ccmc.gsfc.nasa.gov/pub/modelweb/atmospheric/msis/nrlmsise00/ (accessed on 24 January 2022). The Dst index was obtained from https://wdc.kugi.kyoto-u.ac.jp/dstae/index.html (accessed on 15 May 2022). The $F_{10.7}$ and Kp indices can be obtained from ftp://ftp.ngdc.noaa.gov/STP/GEOMAGNETIC_DATA/INDICES/KP_AP/ (accessed on 15 May 2022) or upon request from the corresponding author.

**Acknowledgments:** We acknowledge the support and funding from the National Space Science Center, Chinese Academy of Sciences. We appreciate the valuable and constructive suggestions from the reviewers.

**Conflicts of Interest:** The authors declare that they have no conflict of interest.

**Appendix A**

The daily indicator $F_{10.7}$ index is at a wavelength of 10.7 cm (2800 MHz) of the solar radio flux density, also called as the flux [41]. For low solar activities, $F_{10.7}$ can be approximately ~70 sfu (solar flux units). At maximum solar activity, $F_{10.7}$ can be boosted to ~370 sfu. Because continuous long-term datasets of $F_{10.7}$ are available, compared to Mg II, the correlation to the EUV and the solar visible radiation is strong, and the data are highly related to the number of sunspots. Studying and forecasting the characteristics of space weather is the main use of the $F_{10.7}$ index [35]. Thermospheric mass density modeling usually employs the centered 81-day $F_{10.7}$ smoothed value. In orbit propagation, the last 81-day $F_{10.7}$ value can yield a better performance [42].

The Kp index, which was designed to evaluate the geomagnetic and solar wind effects caused by energy inputs, has a resolution of 3 h and is computed based on the maximum value of the horizontal components of the fluctuations of the geomagnetic field. The field strength observed by the magnetometer is distributed at mid-latitudes [43]. More plainly, the Kp index is used to measure variations in the geomagnetic field in the range of 0 to 9 integers.

The Dst index, also called the disturbance storm time, was developed to present the intensity of globally symmetrical equatorial currents (called 'ring currents') during the main phase of geomagnetic storms [35]. The Dst index is measured hourly and computed from the average geomagnetic field observations near the Earth's equator. Thus, this index indiscriminately measures the effects of many ground and magnetospheric current systems. When $30$ nT $\leq |D\text{st}| \leq 50$ nT, a weak geomagnetic storm occurs. At $50$ nT $\leq |D\text{st}| \leq 100$ nT, a moderate geomagnetic storm occurs. At $100$ nT $\leq |D\text{st}|$, an intense geomagnetic storm occurs.

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
