# Peer review of "Analysis of Orbital Atmospheric Density from QQ-Satellite Precision Orbits Based on GNSS Observations"

_remotesensing, doi:10.3390/rs14163873_

Round 1
Reviewer 1 Report
Overall the manuscript is well written with novelty, clear methodology, valid analysis and comprehensive result. Before proceeding to the publication stage, I encourage the authors to address the following questions / comments:
1. Line 147-149: "For this study, several days of POD in situ detection data files from 1 to 8 November 2021 were processed, which was selected because of high geomagnetic activity." Please provide the Kp index values during these days.
2. Equation (1), suggest to use a_drag on the left hand side as the notation of atmospheric drag acceleration. This is to be consistent with Eq. 8-28 in your reference Vallado textbook
3. Figure 3. Please clarify the meaning of each legend. What do Rrms, Trms, and Nrms stand for?
4. Line 185-187: With respect to a spherical satellite, its precise orbital data can be determined to a resolution of a few centimeters with high-precision GNSS receivers from its obtained dense BDS and GPS tracking observations." How is the precise positioning determined using the GNSS receivers? Please specify the GNSS receiver models and positioning modes (e.g., PPP?) Even during the high geomagnetic day (Nov 4, 2021), the positioning accuracy remained high and robust. Readers may wondering the method behind it.
5. Figure 6. I am not clear about the difference between (a) and (6), or between (c) and (d). Please clarify.
6. Figure 11, please increase the font size so that readers can see the plots clearly. Also, Please specify what R, T, N stand for (radial, along-track, cross-track)?
7. Why the NRLMSISE-00 model is chosen as the benchmark in this study among other atmospheric models? As mentioned by the authors, the NRLMSISE-00 model is relatively less sensitive to space weather activity. The following models maybe be considered for your future study: Thermosphere Ionosphere Electrodynamics General Circulation Model (TIE-GCM) model [1] derived neutral density, Coupled Thermosphere Ionosphere Plasmasphere Electrodynamics Model (CTIPe) model, GITM [3], etc.
[1] Peng, Y., Scales, W.A. & Lin, D. GNSS-based hardware-in-the-loop simulations of spacecraft formation flying with the global ionospheric model TIEGCM. GPS Solut 25, 65 (2021). https://doi.org/10.1007/s10291-021-01099-x
[2] Codrescu, M.V., Negrea, C., Fedrizzi, M., Fuller‐Rowell, T.J., Dobin, A., Jakowsky, N., Khalsa, H., Matsuo, T. and Maruyama, N., 2012. A real‐time run of the Coupled Thermosphere Ionosphere Plasmasphere Electrodynamics (CTIPe) model. Space Weather, 10(2). https://doi.org/10.1029/2011SW000736
[3] Ridley, A. J., Y. Deng, and G. Toth., 2006, The Global Ionosphere-Thermosphere Model (GITM). J. Atmos. Solar-Terrestr. Phys. 68, 839-864.Ridley, A. J., Y. Deng, and G. Toth., 2006, The Global Ionosphere-Thermosphere Model (GITM). J. Atmos. Solar-Terrestr. Phys. 68, 839-864. https://doi.org/10.1016/j.jastp.2006.01.008
Reviewer 2 Report
The paper uses the QQ-satellite orbit to calculate the non-gravitational forces, thus the atmospheric density. It is meaningful to the space whether study.
1. The language expressions are rather confusing and I feel very difficult to follow the authors. I highly recommend the whole paper to be revised w.r.t language. The attached pdf has marked and corrected some, but more to be revised.
2. Related to the content:
The derived densities have been compared to those from the NRLMSISE-00 model. The paper lacks the evaluation (i.e., numbers) of difference between them. Both densities are applied to satellite orbit propagation. However, as I understand, the atmospheric drag coefficient is approximately assumed by the authors. The coefficient is kept the same when deriving densities and conducting orbit predication, in this case the bias in the coefficient would not affect orbit accuracy. Whereas, for NRLMSISE-00 model, the coefficient bias will affect the orbit integration accuracy. Then it would be unfair to conclude that the density from POD is better than those from NRLMSISE-00 model. How do you certify this?

Round 2
Reviewer 2 Report
Thanks for the reply and modification of the language. However, I find the expression is still not professional and quite confusing at some points. It cannot meet the standard of an academic paper. I highly suggest the author read some papers in the related field and use professional expressions, also shorten the sentences to make it easier to understand.
Author Response
Please see the attachment.

This manuscript is a resubmission of an earlier submission. The following is a list of the peer review reports and author responses from that submission.